# Neighborhood environment and incident diabetes, a neighborhood environment-wide association study ('NE-WAS'): Results from the Hispanic Community Health Study/Study of Latinos (HCHS/SOL)

Cara M. Smith [1]*, Elizabeth W. Spalt [1], Linda C. Gallo [2], Jordan Carlson[3], Matthew Allison[4], Daniela Sotres-Alvarez[5], Christina Cordero[6], Qibin Qi[7], Earle C. Chambers[8], Martha Daviglus[9], Amber Pirzada[9], Gregory A. Talavera[2], Robert Kaplan[7], Joel D. Kaufman[1], Stephen J. Mooney[10]

1 Department of Environmental and Occupational Health Sciences, School of Public Health, University of Washington, Seattle, Washington, United States of America, 2 Department of Psychology, San Diego State University, San Diego, California, United States of America, 3 Department of Pediatrics, Children's Mercy Hospital and University of Missouri, Kansas City, Kansas City, Missouri, United States of America, 4 Department of Family Medicine, University of California San Diego, La Jolla, California, United States of America, 5 Collaborative Studies Coordinating Center, Department of Biostatistics, University of North Carolina at Chapel Hill, Chapel Hill, North Carolina, United States of America, 6 Department of Psychology and Behavioral Medicine Research Center, University of Miami, Miami, Florida, United States of America, 7 Department of Epidemiology and Population Health, Albert Einstein College of Medicine, Bronx, New York, United States of America, 8 Department of Family and Social Medicine, Albert Einstein College of Medicine, Bronx, New York, United States of America, 9 Institute for Minority Health Research, University of Illinois at Chicago, Chicago, Illinois, United States of America, 10 Department of Epidemiology, School of Public Health, University of Washington, Seattle Washington, United States of America

* cara15@uw.edu

## Abstract

The prevalence of type 2 diabetes is increasing among the Hispanic/Latino population. Type 2 diabetes incidence rates vary between neighborhoods, but no single aspect of the neighborhood environment is known to cause type 2 diabetes. Using data from the Hispanic Community Health Study/Study of Latinos cohort of 16,415 Hispanic/Latino adults in four major US cities, we conducted a neighborhood environment-wide association study to identify neighborhood measures or clusters of measures associated with diabetes incidence. Two-hundred and four neighborhood measures were calculated at the census tract level or within a 1-km buffer of participants' residential addresses. Independent covariate-adjusted and survey-weighted Poisson regressions were run for each neighborhood measure and incident diabetes. Principal component analysis of neighborhood measures was conducted to reduce dimensionality. No coherent pattern of neighborhood measures or principal component scores were associated with diabetes incidence within the cohort, though established individual-level risk factors such as age and family history were strongly associated with diabetes incidence. Results from our analysis did not indicate specific

**Data availability statement:** The Hispanic Community Health Study / Study of Latinos (HCHS/SOL) is a multi-center epidemiologic study supported by contracts with the National Heart, Lung, and Blood Institute (NHLBI). Due to the data restrictions imposed by the governing IRBs that oversee this human subject research, data access in HCHS/SOL is limited. The data used for this manuscript contained the location of their residents which is identifiable information that cannot be shared without an agreement with the governing IRBs. The data for this study can be accessed by request through the HCHS/SOL study website, https://sites.cscc.unc.edu/hchs/New%20 Investigator%20Opportunities. If you have questions about accessing the data, please feel free to send an email to HCHSAdministration@ unc.edu.

**Funding:** This work was supported by the National Institute of Environmental Health Sciences [R01ES030994: SOLAir: Environmental Factors and Diabetes Development in Latinos]; And the National Institute of Health [5P30ES007033: Interdisciplinary Center for Exposures, Diseases, Genomics, and Environment (EDGE Center)]. Awards received by JK. https://www. niehs.nih.gov/funding/grants, https://www.nih. gov/grants-funding The Hispanic Community Health Study/Study of Latinos is a collaborative study supported by contracts from the National Heart, Lung, and Blood Institute (NHLBI) to the University of North Carolina (HHSN268201300001I / N01-HC-65233), University of Miami (HHSN268201300004I/ N01-HC-65234), Albert Einstein College of Medicine (HHSN268201300002I / N01-HC-65235), University of Illinois at Chicago (HHSN268201300003I / N01- HC-65236 Northwestern Univ), and San Diego State University (HHSN268201300005I / N01-HC-65237). The following Institutes/Centers/ Offices have contributed to the HCHS/SOL through a transfer of funds to the NHLBI: National Institute on Minority Health and Health Disparities, National Institute on Deafness and Other Communication Disorders, National Institute of Dental and Craniofacial Research, National Institute of Diabetes and Digestive and Kidney Diseases, National Institute of

neighborhood measures, clusters, or patterns. Individual, rather than neighborhood, factors distinguish incident diabetes cases from non-cases.

## Introduction

Type 2 diabetes (T2D) prevalence is steadily increasing globally [1]. In the United States, there is particular concern regarding T2D among Hispanic/Latino adults as they are 70% more likely to be diagnosed with T2D than non-Hispanic White adults [2], and are more likely to develop T2D at younger ages [3,4]. Established risk factors of T2D include genetic factors, age, obesity, physical activity, and diet. Natural, built, and social environmental factors may also affect the risk of T2D by influencing other risk factors such as physical activity patterns and diet, or by altering physiological stress responses [5–9].

The neighborhood environment can both promote and discourage physical activity. Sidewalks, bike lanes, connected streets, and nearby businesses can help promote walking and active transportation [10,11]. Lack of sidewalks or sidewalks in poor condition can discourage physical activity [12,13]. In a cross-sectional analysis of San Diego based participants of the Hispanic Community Health Study/Study of Latinos (HCHS/SOL), those areas with higher residential density, retail density, or recreation areas reported higher active transportation than those in lower areas [14].

Neighborhood environment characteristics have also been linked to T2D. In the San Diego site of the HCHS/SOL, greater neighborhood socioeconomic deprivation was found to be associated with higher HbA1c levels (Coeff: 0.08, 95% CI: 0.04, 0.12) and having worse diabetes status at baseline (OR: 1.25, 95% CI: 1.06, 1.47) [15]. Greater neighborhood socioeconomic deprivation was also associated with worsening diabetes status at the second visit (OR: 1.27, 95% CI: 1.10, 1.46) [15]. Dendup et al. (2021) found lack of access to local amenities can increase risk of T2D through associations with higher body mass index (BMI) and lower physical activity. A longitudinal study of Black adults in Jackson, Mississippi found neighborhood problems (noise, heavy traffic, speeding cars, lack of access to adequate food or shopping, litter) to be associated with incident T2D (1.24, 95% CI: 1.00, 1.54, respectively) [16]. This study also found physical activity to be a potential mediator between neighborhood violence and problems and the prevalence of T2D [16]. Lack of access to healthy foods and greater proximity to fast-food establishments can also encourage diets known to be associated with increased T2D risk [17,18].

In aggregate, these studies suggest that neighborhood environment factors can affect T2D risk. However, some of these factors, such as heavy traffic and noise, may be highly correlated, and high levels of inter-correlation make it challenging to isolate the impact of specific environmental factors on T2D risk. Additionally, previous inclusion of specific neighborhood environmental factors in epidemiological analysis of T2D has been largely based on *a priori* hypotheses and all potential factors have not been evaluated agnostically. While having an *a priori* hypothesis is a strength and recommended, literature on neighborhood environment measures and T2D is limited and important measures or classes of measures may have been overlooked due to

Neurological Disorders and Stroke, NIH Institution-Office of Dietary Supplements. https://www.nhlbi.nih.gov/grants-and-training/funding-opportunities-and-contacts This publication was developed under a STAR research assistance agreements, No. RD831697 (MESA Air) and RD-83830001 (MESA Air Next Stage), awarded by the U.S Environmental Protection Agency. It has not been formally reviewed by the EPA. The views expressed in this document are solely those of the authors and the EPA does not endorse any products or commercial services mentioned in this publication. Awards received by JK. https://www.federalgrantswire.com/science-to-achieve-results-star-program.html The funders had no role in study design, data collection and analysis, decision to publish, or preparation of the manuscript.

**Competing interests:** The authors have declared that no competing interests exist.

lack of prior investigation. Therefore, in the analysis presented below, we adopted a hypothesis-generating 'big data' approach inspired by genome-wide association studies (GWAS) to uncover patterns in associations of environmental factors with T2D risk. Our goal was to compare these associations and identify the factors most likely to benefit from future hypothesis-driven studies. This approach, termed "Neighborhood environment-wide association study ('NE-WAS')" is an empirical way to systematically identify neighborhood environment factors associated with an outcome of interest [19–21]. This NE-WAS investigated the association between hundreds of neighborhood environment variables linked to residential address and incident diabetes. We aimed to identify neighborhood-scale environment factors most strongly associated with incident diabetes in this population.

## Methods

### Study design and population

We used data from the Hispanic Community Health Study/Study of Latinos (HCHS/SOL). HCHS/SOL is a longitudinal cohort made up of 16,415 self-identifying Hispanic/Latinos adults at least 18 years in age at enrollment from four US cities [22]. More details on the HCHS/SOL sample and study design have been previously reported [23,24]. Briefly, the cohort enrollment and in-person baseline visits spanned from 2008 to 2011 in the Bronx, Chicago, Miami, and San Diego. Yearly follow-up phone calls were conducted and a second round of in-person visits (visit 2) occurred between 2014 and 2017.

The in-person visits at baseline and visit 2 included questionnaires and clinical examinations. The questionnaires assessed demographics, current health and medical history, and socioeconomic status [24]. Clinical examinations included collection of blood and urine for assays of cardiovascular risk factors [24]. The yearly follow-up phone calls contained questions on general health, potential updates from their doctors or health professional, and any potential hospitalization or emergency room visits that occurred since the last follow-up [24]. The study population for this analysis included participants who completed the baseline and 2nd visit and were free of diabetes at baseline. All participating institutions received approval by their respective institutional review boards (Albert Einstein College of Medicine, San Diego State University, University of Illinois at Chicago, University of Miami, and University of North Carolina) and written informed consent, in either English or Spanish, was received from all participants. Study #00010745 was also approved by the University of Washington IRB. Data for the HCHS/SOL cohort was accessed on February 17th 2023.

### Diabetes ascertainment

Diabetes status was collected at baseline, during each follow-up phone call, and at the second in-person visit. At these times, participants were specifically asked if a doctor or health care professional has told them they have diabetes or high sugar in their blood and if they are receiving any treatment [24]. During the in-person visits, blood tests were administered to test fasting plasma glucose (FPG) and glycosylated

hemoglobin (HbA1c) [24]. A 2-hour oral glucose tolerance test (OGTT) was also conducted on participants with who did not self-report a diabetes diagnosis and on participants with FPG > 150 mg/dL [25]. Based on the American Diabetes Association criteria and data collection practices, we defined diabetes in this analysis in two ways. Our primary definition was having an FPG ≥ 126 mg/dL; HbA1c level ≥6.5%; post-OGTT glucose ≥ 200 mg/dL; and/or use of antihyperglycemic drugs. Our secondary definition included all those who fit the primary definition of diabetes and additionally includes those who self-reported diabetes or high sugar in their blood. Ascertainment of diabetes in the HCHS/SOL cohort was not able to distinguish between type 1 and type 2 diabetes.

## Neighborhood environment measures

Participant residential addresses at baseline were collected and geocoded. Of the 16,415 enrolled participants, 16,390 had their baseline residential addresses geocoded. There were 8,288 unique baseline addresses as multiple participants could live in the same household. For each residential address, we compiled a data set of 361 neighborhood environment measures. These neighborhood environmental measures were collected from publicly available data sources including the 2010 Census and American Community Survey, Bureau of Transportation Statistics, National Emission Inventory data-base and the National Neighborhood Data archive (NaNDA). This archive includes data from the National Transit Map, the National Establishment Time Series database, and the Federal Communications Commission. Data from these sources were accessed and downloaded on February 17th 2023.

We removed 31 neighborhood measures related to the amount and type of schools in a census tract due to high percentage of missingness. We also removed measures with standard deviation of zero, and measures where more than 90% of participants had a zero value for said measure from the dataset (e.g., confined feeding operations land use in 1 km buffer).

We computed Pearson correlation coefficients for all pairs of neighborhood measures and excluded one measure of any pair whose coefficient exceeded 0.9. For example, the percentage of households with a smartphone or tablet and the percentage of households with any computing device were correlated at 0.94. Accordingly, we removed the percentage of households with a smartphone or tablet from our measure list. For each of the remaining 204 measures, we generated histograms to observe distribution for both non-transformed and log-transformed data. For 58 (28.4%) of these measures, the log-transformed data distribution visually appeared more normal than the non-transformed measures. For these measures (detailed in S1 Table), we used log-transformed measures in analysis; for the other 146, we used the non-transformed version.

A list of all neighborhood environment measures and the data sources used in analysis can be found in S1 Table. Forty-two measures (concerning features including length of A3 roads, elevations above sea level, and deciduous forests) were measured within a 1 km radius buffer of the home. One-hundred and sixty-two measures (concerning features including percentage occupied housing units occupied by renter, density of social services organizations, and proportion of open park land) were measured within the 2010 census tract.

## Inclusion criteria

Starting with the 11,619 participants with ascertainment of diabetes status at visit 2, analysis was limited to those who were free of diabetes at baseline (N = 9209 for primary diabetes outcome and N = 9,077 for secondary diabetes outcome). Analysis was further limited to participants with a successfully geocoded residential address (N = 8006 for primary diabetes outcome and N = 7889 for secondary diabetes outcome).

## Statistical analysis

For the primary analysis, we ran individual Poisson regression models for incident diabetes at visit 2 and each neighborhood environment variable z-score controlling for age and gender. We used years in the study as an offset to allow for

different amounts of time at risk between participants. Our secondary analysis adjusted for additional person-level covariates that tend to confound environmental exposure hypotheses: educational attainment, income, marital status, duration of US residence, family history of diabetes, Hispanic/Latino heritage, and study center. For all regression models we used complete case analysis. We applied sampling weights to account for the HCHS/SOL complex sampling design, those living in the same household, and participant non-response at visit 2 by using the *survey* package in R [23]. The sampling weights were also calibrated to the age, sex, and Hispanic/Latino heritage from the 2010 US Census. To account for multiple comparisons, we used the Benjamini-Hochberg false discovery rate (FDR) approach.

Manhattan plots were generated to visually inspect patterns of variable categories and their strength of association with incident diabetes. For these plots, neighborhood measures were categorized into one of six bins: 'Demographics and Households', 'Education, Employment, and Income', 'Goods and Services', 'Healthcare', 'Recreation and Transportation', and 'Urban Form' by expert opinion. These groupings were to help with visual interpretation of the plots and not involved in the modeling. The bin categorization for each measure is included in S1 Table. Then, consistent with the approach used to visualize GWAS results, we plotted the negative log p-value (representing strength of association) for each measure and ordered them in categorical groups.

As an alternate approach for handling dependence between the 204 neighborhood measures and to reduce dimensionality, we conducted a Principal Component Analysis (PCA). We initially ran this PCA with the same data set used in the individual measure Poisson regression models. Results showed strong influence by study center. To identify correlation of components irrespective of study center, we regressed all the neighborhood measures on study center and conducted a PCA of the residuals. We then used the first and second components, which accounted for 16% of the total variance, as predictors in a Poisson regression model adjusting for age, sex, educational attainment, income, marital status, duration of US residence, family history of diabetes, and Hispanic/Latino heritage group.

### Sensitivity analysis

In a sensitivity analysis, we repeated analyses using the diabetes outcome definition that included self-reported diagnosis. As the exposures are only measured at residential baseline addresses and participants may move to a different location during follow-up with different exposure levels, we also conducted a sensitivity analysis on the subset of participants (n = 4188; 52.3% of total) who did not move during follow-up.

All analyses were conducted in R Studio (Ver 4.1.3)

### Results

Descriptive statistics of the 8,006 study participants included in the primary analysis are presented in Table 1. A descriptive table by study center is presented in S2 Table. The weighted average age was 39.2 years old and the weighted percentage of females was 50.5% (Table 1). The age-adjusted diabetes incidence rate between baseline and visit 2 for the analytic cohort was 16.53 cases per 1,000 person-years. There were 194 incident cases in The Bronx, 283 in Chicago, 185 in Miami, and 264 in San Diego.

The NE-WAS independent Poisson regression models on the association between the primary diabetes outcome and 204 neighborhood measurements, adjusted for age and sex, yielded 37 statistically significant associations at $p < 0.05$; however, none of these remained statistically significant at the FDR-adjusted $p < 0.05$ level. The fully adjusted model yielded 11 statistically significant associations; none of which were statistically significant at the FDR-adjusted $p < 0.05$ level.

Center-specific NE-WAS independent Poisson regressions adjusting for age and sex yielded one neighborhood measure significant at the FDR-adjusted $p < 0.05$ level in Chicago, 0 in the Bronx, 7 in Miami, and 0 in San Diego (Table 2). Manhattan plots for each study center are presented in Fig 1.

**Table 1. Characteristics of participants included in analysis of primary diabetes outcome (N = 8006).**

| Variables | Sample Weighted % or Mean (SD) |
|---|---|
| **Female** | 50.5 |
| **Age, years** | 39.2 (13.9) |
| **Waist Circumference (cm)** | 96.3 (13.7) |
| **Hispanic/Latino Heritage** | |
| Central American | 7.2 |
| Cuban | 18.2 |
| Dominican | 8.4 |
| Mexican | 40.9 |
| Puerto Rican | 14.5 |
| South American | 5.2 |
| More than one heritage | 3.8 |
| Other | 0.8 |
| **Years in the US** | |
| Less than 10 years | 27.8 |
| 10 years or more | 47.8 |
| US Born | 24.4 |
| **Family History of Diabetes** | 37.5 |
| **Marital Status** | |
| Single | 35.1 |
| Married or living with a Partner | 50.1 |
| Separated, Divorced, or Widow | 14.8 |
| **Education** | |
| No High School Diploma or GED | 28.1 |
| At most a High school diploma or GED | 29.2 |
| High school (or GED) education | 13.4 |
| University/college education | 29.3 |
| **Income** | |
| Less than $10,000 | 13.7 |
| $10,001-$20,000 | 31.5 |
| $20,001-$40,000 | 33.8 |
| $40,001-$75,000 | 15.3 |
| More than $75,000 | 5.8 |
| **Years between Visit 1 and Visit 2** | 6.14 (0.86) |
| **Study Center** | |
| Bronx | 25.5 |
| Chicago | 17.4 |
| Miami | 28.3 |
| San Diego | 28.8 |
| **Did not move between Baseline and Visit 2** | 47.4 |
| **Visit 2 Diabetes Status** | |
| Diabetes (blood test and self-report medication | 9.13 |
| Diabetes (blood test, self-report medication, and self-reported diagnosis) | 13.5 |

**Table 2. Neighborhood measures significantly associated with incident diabetes with an FDR adjustment at the *p*<0.05 level in age and sex adjusted, center-specific models.**

| Measure | Bin | Center | Incident Cases | IRR – Age and Sex Adjusted |
|---|---|---|---|---|
| **Developed open space in 1 km radius raster*** | Urban Form | Chicago | 283 | 1.19 |
| **Number of Physical, Occupational, and Speech Therapists per square mile with 2 + employees** | Healthcare | Miami | 185 | 1.17 |
| **Percentage of households with laptop/desktop** | Demographics and Households | Miami | 185 | 0.77 |
| **Tract level population for whom poverty status is determined under 1.00** | Education, Employment, and Income | Miami | 185 | 1.26 |
| **Tract level median value for specified owner-occupied housing units** | Demographics and Households | Miami | 185 | 0.77 |
| **Tract level percentage of households with income less than $25,000** | Education, Employment, and Income | Miami | 185 | 1.29 |
| **Tract level percentage of households with income less than $100,000** | Education, Employment, and Income | Miami | 185 | 1.62 |
| **Tract level percentage of households with income less than $150,000** | Education, Employment, and Income | Miami | 185 | 2.09 |

Note: IRR – Incidence Rate Ratio. Z-scores of neighborhood measures used in analysis.

*Measure was log-transformed for analysis.

The first and second principal components (PC) of the PCA explained 9.6% and 6.8% of the total variance, respectively (S3 Fig). These component scores were not associated with diabetes incidence. The IRR for the first component was 1.01 (95% CI: 0.99–1.03), and IRR for the second component was 1.01 (95% CI: 0.99–1.03). The most important predictors of diabetes in all models were age and family history of diabetes, which remained significantly associated with diabetes incidence even after adjustment for neighborhood factors (S4 Table).

In analyses expanding the diabetes incidence definition to include self-reported diagnosis, the age-adjusted incidence rate was 24.4 cases/1,000 person-years. Associations for the fully adjusted and study center specific models were similar. However, for the age and sex adjusted model, 61 exposures were statistically significant and 20 were statistically significant after FDR correction (Table 3).

The population who did not move differed from the population that did move by a number of demographics including age, years in the US, and family history of diabetes (S5 Table). Restricting the analysis to those participants who did not move during follow-up (52.3%) yielded similar results for the fully adjusted model and the San Diego model when using the primary definition of diabetes. However, for the model only adjusting for age and sex, 11 neighborhood measures were significant with the FDR correction (Table 4). For the center-specific models, Chicago again only had one significant measure, number of full-service restaurants per square mile with 2 or more employees, with the FDR correction. The Bronx had two significant measures with the FDR correction, the number of home health services per 1,000 people with 2 + employees and the density of furniture stores in 2010. Miami only had one significant measure with the FDR correction, meters to medium port.

## Discussion

Using an empirical high-dimensional analysis approach, we assessed the association between 204 neighborhood environment measures and diabetes incidence in a Hispanic/Latino cohort. Neither the fully adjusted nor the age and sex adjusted NE-WAS independent Poisson regressions yielded any significant associations between neighborhood measures and diabetes incidence. While a few neighborhood measures were associated with diabetes incidence in center-specific models, these measures varied across cities. Taken together, these results suggest that, if there is a strong environmental determinant of diabetes, that determinant is largely uncorrelated with commonly measured neighborhood features.

Our sensitivity analysis including self-reported diabetes in the outcome assessment resulted in more neighborhood measures associated with diabetes. Including the self-reported diabetes increased the age-adjusted incidence

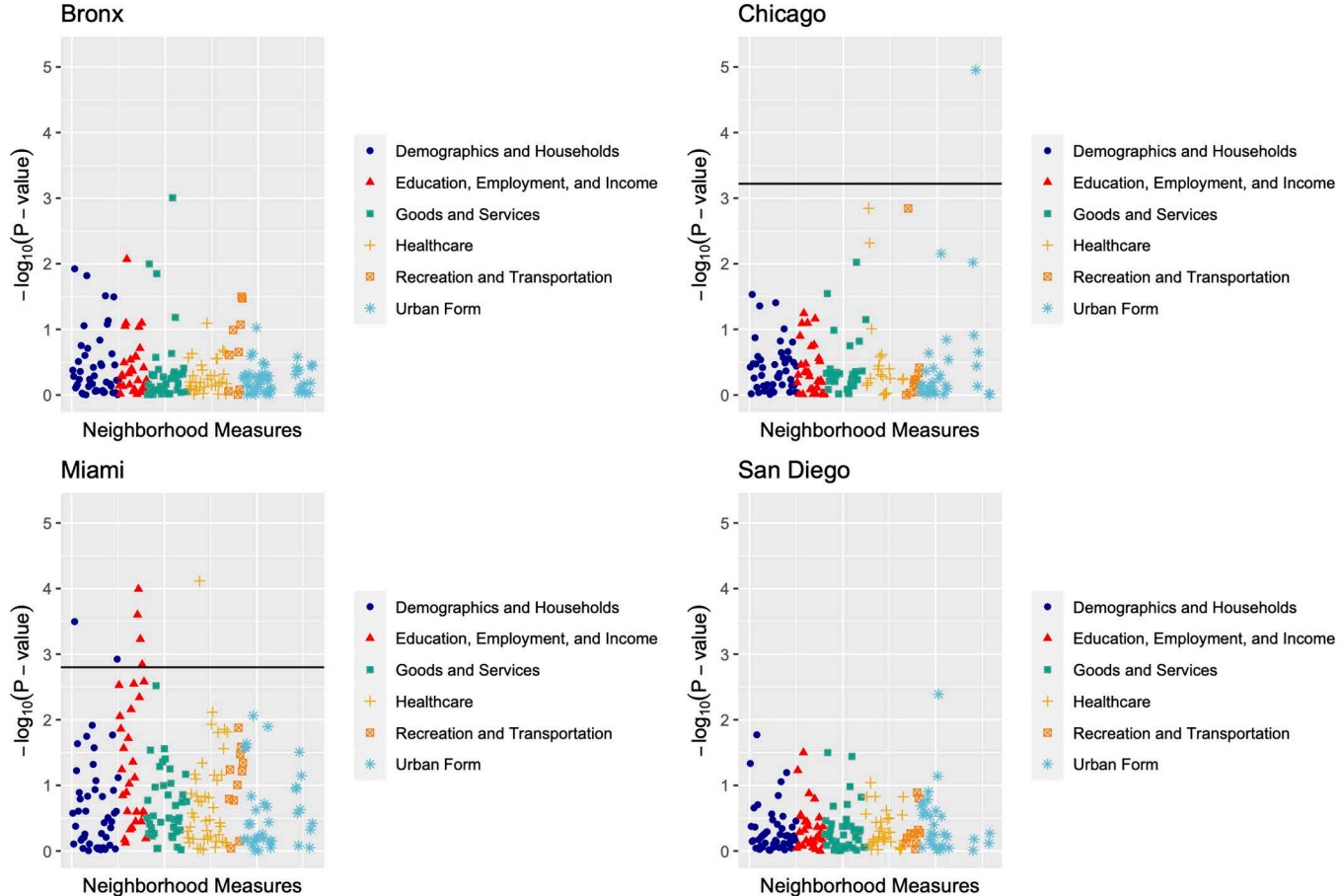

**Fig 1. Manhattan plots of the negative log p-values from the independent Poisson regressions of 204 neighborhood measures and incident type 2 diabetes adjusting for age and sex in each study center.** FDR and Bonferroni corrections lines are in black. For Chicago, the FDR threshold is 3.22, and for Miami the FDR threshold is 2.80.

rate, which may explain the increase in associated neighborhood measures as the two samples were statistically different based on demographic characteristics such as age, waist circumference, sex, and Hispanic/Latino heritage (S6 Table). Our second sensitivity analysis on those who did not move during the follow-up period also resulted in more associated neighborhood measures than the primary analysis. Differences on several diabetes risk factors were observed between those participants who did and did move between visits (S5 Table). This indicates that those who move are not a random subsample and therefore provided a rationale for adjusting the statistical models for moving status. There is little overlap between the sets of statistically significant neighborhood measures from our two sensitivity analyses, which are presented in Table 3 and Table 4. While our sensitivity analysis showed how different samples of the study population can vary the results, we overall have little confidence these results indicate any causal relationship.

Neither the NE-WAS nor the PCA results identified a clearly interpretable set of neighborhood measurements or clusters of measurements. Individual risk factors such as age and family history of diabetes appear to matter more than the neighborhood environment risk factors included in this analysis. This is consistent with well-established individual risk factors of T2D [1,2,26,27]. Additionally, while investigation of effect modification of neighborhood measures on the

**Table 3. Neighborhood measures significantly associated with incident diabetes (including self-report) with an FDR adjustment at the *p*<0.05 level in age and sex adjusted models.**

| Measure | Bin | IRR – Age and Sex Adjusted |
|---|---|---|
| Density of civic and social organization in 2010 (count/total pop) | Goods and Services | 0.87 |
| Density of performing arts companies in 2010 (count/pop) | Recreation and Transportation | 0.85 |
| Developed low intensity in 1 km radius raster* | Urban Form | 0.87 |
| Elevation above sea level in meter* | Urban Form | 1.18 |
| Meters to intersections of A2 and A3 roads | Urban Form | 1.12 |
| Meters to large airport | Urban Form | 1.11 |
| Meters to medium port | Urban Form | 0.87 |
| Meters to a small port | Urban Form | 1.17 |
| Number of nursing and residential care facilities per 1000 people with 2+employees | Healthcare | 0.89 |
| Number of other health practitioners per sq. mile with 2+employees | Healthcare | 0.89 |
| Number of physicians (expect mental health) per 1000 people with 2+employees | Healthcare | 0.89 |
| Standard deviation of elevation of twenty points surrounding location* | Urban Form | 1.16 |
| Tract level % aged 5–9 | Demographics and Households | 1.14 |
| Tract level % aged 10–14 | Demographics and Households | 1.11 |
| Tract level % aged 15–17 | Demographics and Households | 1.12 |
| Tract level % aged 65–75* | Demographics and Households | 0.89 |
| Tract level % aged 85+ | Demographics and Households | 0.88 |
| Tract level % foreign born | Demographics and Households | 0.90 |
| Tract level % occupied housing units using electric for heating fuel | Demographics and Households | 0.90 |
| Tract level % occupied housing units using gas for heating fuel | Demographics and Households | 1.17 |

Note: IRR – Incidence Rate Ratio. Z-scores of neighborhood measures used in analysis.

*Measure was log-transformed for analysis

relationship between individual risk factors and diabetes incidence was out of the scope of this manuscript, it is an important next step for future studies.

Contrary to our results, previous research has linked neighborhood environment and neighborhood socioeconomic constructs to T2D [15,16,28]. In the San Diego site of the HCHS/SOL, greater neighborhood socioeconomic deprivation was associated with worsening diabetes status (OR: 1.27, 95% CI: 1.10, 1.46) after 6 years of follow-up [15]. Previous studies on neighborhood environment and incident T2D clustered related neighborhood measures into a single variable such as "neighborhood problems", "neighborhood violence", "social disorder", or "neighborhood socioeconomic deprivation" using principal component analysis [15,16]. These designs of purposefully clustering measures into a single metric may be a possible reason for contrary results. However, our PCA did not yield any clusters explaining large proportions of the variation, and all components were null in prediction of diabetes.

Strengths of this analysis included the utilization of a large well-established cohort of self-identifying Hispanic/Latino adults, an underrepresented group in diabetes research, in 4 major US metropolitan areas. Participants were from a range

**Table 4. Neighborhood measures significantly associated with incident diabetes with an FDR adjustment at the *p*<0.05 level in age and sex adjusted models among those who did not move between baseline and follow-up.**

| Measure | Bin | IRR – Age and Sex Adjusted |
|---|---|---|
| Count of transit stops reported to National Transit Map as of 2018* | Urban Form | 0.84 |
| Number of full-service restaurants per sq. mile with 2+employees* | Goods and Services | 1.30 |
| Number of supermarkets/grocery stores per 1000 people with 2+employees* | Goods and Services | 1.27 |
| Population in a 1 km buffer | Demographics and Households | 1.27 |
| Tract level percent employed civilian population 16 years and over in wholesale trade* | Education, Employment, and Income | 0.81 |
| Tract level percent occupied housing units using electricity for heating fuel* | Demographics and Households | 0.81 |
| Tract level percent occupied housing units using oil for heating fuel | Demographics and Households | 1.20 |
| Tract level percent of housing units with 1 unit* | Demographics and Households | 0.84 |
| Tract level percent population 15 years and over never married | Demographics and Households | 1.20 |
| Tract level percent race, White alone | Demographics and Households | 0.82 |
| Tract level percent race, other | Demographics and Households | 1.27 |

Note: IRR – Incidence Rate Ratio. Z-scores of neighborhood measures used in analysis.

*Measure was log-transformed for analysis

of Hispanic/Latino heritage groups. This analysis allowed for the investigation of the relationship between 204 neighborhood measures and incident diabetes.

There are a few limitations to note. First, while the NE-WAS analysis is explicitly exploratory, this exploration relies on an assumption that the most promising measures for follow-up study are those with strong unadjusted associations with diabetes on a direct pathway between exposure and diabetes. While we adjusted for a set of individual covariates, it is possible there may be environmental or neighborhood level confounding that weakens associations between a truly important neighborhood measure and diabetes incidence. Our concerns on this count are somewhat mitigated by our removal of tightly correlated pairs of neighborhood measures and the robustness of our null findings to different specifications of diabetes. Second, unlike in GWAS analysis, our neighborhood measures are continuous and can be untransformed or log-transformed. Our choice of transformations was principled but may have result dose-response patterns that are artifacts of our choices.

Third, while we examined 204 different measures, those measures were selected for convenience based on their availability: neighborhood measures that were not readily available to us, such as traffic density and noise [16,29–31], may have stronger associations with T2D. Fourth, our selection of a 1 km buffer for the participant's neighborhood may have precluded identifying associations that might exist with measures at other buffer sizes. While a 1-km buffer is commonly used for neighborhood environments, other buffer distances may be relevant and important [19,32]. However, some of our neighborhood measures were reported at the census tract level, and would remain unchanged with varying neighborhood buffers. Yet, the neighborhood environment outside their 1-km residential area or census tract, such as their place of work and larger community environment, was unaccounted for and may affect T2D risk as well.

Fifth, the average follow-up time was 6.14 years which may not be of sufficient duration to develop T2D. As the appropriate exposure window for each exposure is unclear, and we do not have residential data prior to baseline, we focused

our analysis on exposure over the follow-up period to investigate diabetes incidence. Additionally, we are unable to distinguish between type 1 and type 2 diabetes in our ascertainment. However, given the age of the HCHS/SOL cohort, the greater majority of incident cases are likely type 2. Lastly, this analysis was focused on 4 major metropolitan areas with large Hispanic/Latino populations in the United States. Results might be different in regions outside the United States or in smaller urban, suburban, or rural areas. Future studies should expand to other cities and regions to increase generalizability.

## Conclusion

We conducted a NE-WAS analysis and PCA to identify specific neighborhood measures, or clusters of measures, associated with diabetes. Overall, results from our analysis did not indicate specific neighborhood measures, clusters, or patterns. Individual risk factors remain better potential intervention targets for type 2 Diabetes. Future non-cross-sectional studies on the HCHS/SOL cohort should investigate the potentially varying associations among movers and non-movers if the exposure is only accounted for at baseline. Future studies can also incorporate data from additional visits of HCHS/SOL.

## Supporting information

**S1 Table. List of neighborhood measures used in analysis, if they were log transformed for analysis, their bin categorization, and their data source.**
(DOCX)

**S2 Table. Characteristics of participants included in analysis of primary diabetes outcome (N = 8006) by study center.**
(DOCX)

**S3 Fig. Scree plot of first 10 dimensions (principal components) from PCA using study center based residuals.**
(TIF)

**S4 Table. Incidence rate ratio (IRR) from Poisson regression of the first two principal components and incident diabetes adjusting for age, sex, education, income, marital status, ethnic background, years in US, family history of diabetes and accounting for survey weights and design.**
(DOCX)

**S5 Table. Characteristics of participants who did not move during follow up (N = 4188) and those who did move (N = 3818).**
(DOCX)

**S6 Table. Characteristics of participants with incident diabetes based on the primary (N = 926) and secondary definition (N = 1323).**
(DOCX)

## Acknowledgments

The authors thank the staff and participants of HCHS/SOL for their important contributions. We also like to thank all those who worked on the National Neighborhood Data Archive (NaNDA); the open data repository was very useful when compiling neighborhood environment data.

## Author contributions

**Conceptualization:** Stephen J. Mooney.

**Data curation:** Cara M. Smith, Elizabeth W. Spalt.

**Formal analysis:** Cara M. Smith, Daniela Sotres-Alvarez, Stephen J. Mooney.

**Funding acquisition:** Linda C. Gallo, Daniela Sotres-Alvarez, Christina Cordero, Earle C. Chambers, Martha Daviglus, Amber Pirzada, Gregory A. Talavera, Robert Kaplan, Joel D. Kaufman.

**Methodology:** Cara M. Smith, Stephen J. Mooney.

**Supervision:** Elizabeth W. Spalt, Robert Kaplan, Joel D. Kaufman, Stephen J. Mooney.

**Writing – original draft:** Cara M. Smith.

**Writing – review & editing:** Cara M. Smith, Elizabeth W. Spalt, Linda C. Gallo, Jordan Carlson, Matthew Allison, Daniela Sotres-Alvarez, Christina Cordero, Qibin Qi, Earle C. Chambers, Gregory A. Talavera, Robert Kaplan, Joel D. Kaufman, Stephen J. Mooney.

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
