## [Decision Letter · Decision Letter 0]

Dear Dr. Smith,

Thank you for submitting your manuscript to PLOS ONE. After careful consideration, we feel that it has merit but does not fully meet PLOS ONE’s publication criteria as it currently stands. Therefore, we invite you to submit a revised version of the manuscript that addresses the points raised during the review process.

We look forward to receiving your revised manuscript.

Kind regards,

Jagadeesh Puvvula, PhD, PharmD, MPH

Academic Editor

PLOS ONE

“This work was supported by the National Institute of Environmental Health Sciences

[R01ES030994: SOLAir: Environmental Factors and Diabetes Development in Latinos]; And the

National Institute of Health [5P30ES007033: Interdisciplinary Center for Exposures, Diseases,

Genomics, and Environment (EDGE Center)]. Awards received by JK.

https://www.niehs.nih.gov/funding/grants

https://www.nih.gov/grants-funding

The Hispanic Community Health Study/Study of Latinos is a collaborative study supported by

contracts from the National Heart, Lung, and Blood Institute (NHLBI) to the University of North

Carolina (HHSN268201300001I / N01-HC-65233), University of Miami (HHSN268201300004I/N01-HC-65234), Albert Einstein College of Medicine (HHSN268201300002I / N01-HC-65235), University of Illinois at Chicago (HHSN268201300003I / N01- HC-65236 Northwestern Univ), and San Diego State University (HHSN268201300005I / N01-HC-65237). The following Institutes/Centers/Offices have contributed to the HCHS/SOL through a transfer of funds to the NHLBI: National Institute on Minority Health and Health Disparities, National Institute on Deafness and Other Communication Disorders, National Institute of Dental and Craniofacial Research, National Institute of Diabetes and Digestive and Kidney Diseases, National Institute of Neurological Disorders and Stroke, NIH Institution-Office of Dietary Supplements.

https://www.nhlbi.nih.gov/grants-and-training/funding-opportunities-and-contacts

This publication was developed under a STAR research assistance agreements, No. RD831697

(MESA Air) and RD-83830001 (MESA Air Next Stage), awarded by the U.S Environmental

Protection Agency. It has not been formally reviewed by the EPA. The views expressed in this

document are solely those of the authors and the EPA does not endorse any products or

commercial services mentioned in this publication. Awards received by JK.

https://www.federalgrantswire.com/science-to-achieve-results-star-program.html”

Reviewers' comments:

Reviewer's Responses to Questions

**Comments to the Author**

1. Is the manuscript technically sound, and do the data support the conclusions?

Reviewer #1: Yes

Reviewer #2: Yes

Reviewer #3: Yes

2. Has the statistical analysis been performed appropriately and rigorously?

Reviewer #1: Yes

Reviewer #2: Yes

Reviewer #3: Yes

3. Have the authors made all data underlying the findings in their manuscript fully available?

Reviewer #1: Yes

Reviewer #2: Yes

Reviewer #3: Yes

4. Is the manuscript presented in an intelligible fashion and written in standard English?

Reviewer #1: Yes

Reviewer #2: Yes

Reviewer #3: Yes

Reviewer #1: I have reviewed this manuscript thoroughly and can confirm that it is technically sound, with relevant supporting data. It is written in clear and simple language, making it easy to understand. However, there are a few minor grammatical errors, and I recommend that the authors proofread the manuscript and address these issues.

Reviewer #2: Good effort by the authors

The utilization of a large well-established cohort is commendable and also targeting an underrepresented group in diabetes research, in 4 major US metropolitan areas.

line 105-109: ''This NE WAS investigated the association between hundreds of neighborhood environment variables

linked to residential address and incident diabetes approximately 6 years later. We also used factor analysis to identify patterns in features of the same suite of variables as predictors of risk, in the population-based Hispanic Community Health Study / Study of Latinos'' . This sentences is not required as part of the introduction. Rather, it should be found in the methods section.

The weighted average age was 39.2 years old. This suggest most of the participants are younger. what proportion of are adult above 65 years old? It was reported that the IRR was adjusted for age. Was any thought given to the impact of retrospective exposures of some of the the participants in the environment/neighborhood measures?

In Table 3. Neighborhood measures significantly associated with incident T2D (including self report) with an FDR adjustment at the p < 0.05 level in age and sex adjusted models, the Tract level % aged 18 to 64 seemed not represented. Any explanation for this?

Line 306: "Contrary to our results, previous research has linked neighborhood environment and neighborhood socioeconomic constructs to T2D [15,16,28]." In addition to the fact that previous studies on neighborhood

environment and incident T2D clustered related neighborhood measures into a single variable: were those previous study longitudinal? and what was the duration of the study/observations? could this have also contributed to the difference in the results from this and previous studies?

Considering desire to have more impactful effect on a person/population, is this study continuous?

Reviewer #3: The authors use an interesting approach here, applying big data and GWAS principles to evaluate neighborhood environment factors associated with T2D. The statistical analyses are sound and conclusions are supported by the reported results. The main weakness of the paper is a lack of discussion regarding selection bias. I would suggest the authors evaluate what the 'general' Hispanic/Latino population looks like in these cities and nationwide, what fraction of them has T2D, neighborhood environment characteristics of areas with Hispanic/Latino population, and how all of this compares to the study populations in the 4 cities. Without this, the generalization of the results are gravely hampered. Even though individual information regarding risk factors will not be available for this evaluation, population level risk factors may offer suggestions as to whether the conclusions from this report are generalizable and in what parts of country (if any).

Other comments:

1) The authors' discussion of people who 'moved' between baseline and visit 2 was also confusing. I would suggest they clarify this point further.

2) Instead of looking for environmental factors that are as strong or stronger than individual factors, I would suggest the authors consider neighborhood environment factors as potential moderators of the relationship between personal risk factors and T2D. This possibility needs to be addressed.

3) Another key factor that deserves discussion is the level of interaction an individual has with their home environment. For instance, some people may interact more with the neighborhood environmental factors at their workplace than home location. This should be considered in Limitations section, along with other personal factors related to physical/mental health, occupation, lifestyle factors, free time, etc. that may influence how a person interacts with their environment in a way that can meaningfully impact their risk of developing T2D.

**Do you want your identity to be public for this peer review?** For information about this choice, including consent withdrawal, please see our Privacy Policy

Reviewer #1: No

Reviewer #2: No

Reviewer #3: No

---

## [Author Response · Author response to Decision Letter 1]

3 Mar 2025

Dear Reviewers and Academic Editor,

Thank you for your time reviewing and providing valuable feedback to this manuscript. Please see below for our responses to your questions and suggestions.

- We have uploaded the file for the figure as Fig1.tiff

- We have made the title sentence case and changed the affiliation indicators to numbers

- Have removed titles from author list

- Added country to affiliations

- Removed full address of corresponding author

- Made headings all bold 18pt font

- Made all level 2 headings bold 16pt font and sentence case

- Made supporting information titles bold

- Uploading individual files for each supporting information table or figure.

- The funders had no role in study design, data collection and analysis, decision to publish, or preparation of the manuscript. This has been updated in the cover letter

3) If there are ethical or legal restrictions on sharing a de-identified data set, please explain them in detail (e.g., data contain potentially identifying or sensitive patient information, data are owned by a third-party organization, etc.) and who has imposed them (e.g., a Research Ethics Committee or Institutional Review Board, etc.). Please also provide contact information for a data access committee, ethics committee, or other institutional body to which data requests may be sent.

- The Hispanic Community Health Study / Study of Latinos (HCHS/SOL) is a multi-center epidemiologic study supported by contracts with the National Heart, Lung, and Blood Institute (NHLBI). Due to the data restrictions imposed by the governing IRBs that oversee this human subject research, data access in HCHS/SOL is limited. The data used for this manuscript contained the location of their residents which is identifiable information that cannot be shared without an agreement with the governing IRBs. The data for this study can be accessed by request through the HCHS/SOL study website, https://sites.cscc.unc.edu/hchs/New%20Investigator%20Opportunities. If you have questions about accessing the data, please feel free to send an email to HCHSAdministration@unc.edu.

- I have updated this response in the submission portal.

4) Please include your full ethics statement in the ‘Methods’ section of your manuscript file. In your statement, please include the full name of the IRB or ethics committee who approved or waived your study, as well as whether or not you obtained informed written or verbal consent. If consent was waived for your study, please include this information in your statement as well

- We have updated the Methods section to contain the following: “All participating institutions received approval by their respective institutional review boards (Albert Einstein College of Medicine, San Diego State University, University of Illinois at Chicago, University of Miami, and University of North Carolina) and written informed consent, in either English or Spanish, was received from all participants. Study #00010745 was also approved by the University of Washington IRB.”

5) Comments from Reviewer #1: I have reviewed this manuscript thoroughly and can confirm that it is technically sound, with relevant supporting data. It is written in clear and simple language, making it easy to understand. However, there are a few minor grammatical errors, and I recommend that the authors proofread the manuscript and address these issues.

- Thank you for your comments. We apologize for the few grammatical errors. We have proofread the paper again and edited these issues.

6) Comments from Reviewer #2:

a) line 105-109: ''This NE WAS investigated the association between hundreds of neighborhood environment variables linked to residential address and incident diabetes approximately 6 years later. We also used factor analysis to identify patterns in features of the same suite of variables as predictors of risk, in the population-based Hispanic Community Health Study / Study of Latinos'' . This sentences is not required as part of the introduction. Rather, it should be found in the methods section.

- Thank you for your note. We have edited the introduction sentences to focus more on the overall study aim. The last two sentences now read as “This NE-WAS investigated the association between hundreds of neighborhood environment variables linked to residential address and incident diabetes. We aimed to identify neighborhood-scale environment factors most strongly associated with incident diabetes in this population.” [Line 107-110]

b) The weighted average age was 39.2 years old. This suggest most of the participants are younger. what proportion of are adult above 65 years old? It was reported that the IRR was adjusted for age. Was any thought given to the impact of retrospective exposures of some of the participants in the environment/neighborhood measures?

- At baseline, 5.5% of the analytic sample was 65 or older. At visit 2, 15% of the sample was 65 or older. Thank you for your questions on retrospective exposures. It is unclear what the appropriate exposure window is especially since this analysis is examining many exposures. So, it is possible the average 6-year follow-up is not long enough. However, we only included those free of diabetes at baseline and focused on cumulative exposure over the follow-up period for the visit 2 incidence. We are unable to examine exposures before baseline as we do not have any residential data from before enrollment in the study. We have added this as a limitation in our discussion. [Line 348 – 351]

c) In Table 3. Neighborhood measures significantly associated with incident T2D (including self-report) with an FDR adjustment at the p < 0.05 level in age and sex adjusted models, the Tract level % aged 18 to 64 seemed not represented. Any explanation for this?

- Thank you for pointing this out. While it cannot be known exactly why this was not a significant association, it is likely due to there being a high percentage of those aged 18 to 64 in each of the tracts, and not much variability in that measure among the included census tracts. It is possible the other associations have arisen by chance.

d) Line 306: "Contrary to our results, previous research has linked neighborhood environment and neighborhood socioeconomic constructs to T2D [15,16,28]." In addition to the fact that previous studies on neighborhood environment and incident T2D clustered related neighborhood measures into a single variable: were those previous study longitudinal? and what was the duration of the study/observations? could this have also contributed to the difference in the results from this and previous studies?

- Thank you for your questions. Citation 15 linked baseline neighborhood deprivation to worsening diabetes status. This was a longitudinal analysis of about 6 years of follow-up. The study population was those in the San Diego region of the HCHS/SOL. Citation 16 was both a cross-sectional and longitudinal analysis. Cross-sectionally, neighborhood problems were associated with T2D prevalence. Longitudinal, over a median follow-up of 7.3 years, higher density of unfavorable food stores was associated with higher T2D incidence. Citation 28 was a review article highlighting previous longitudinal research. While follow-up periods were similar, it is possible their designed clusters of related neighborhood measures into a single variable contributed to the difference. We have added information about the follow-up times of other studies to the text and included this new sentence: “These designs of purposefully clustering measures into a single metric may be a possible reason for contrary results.” [Line 318 – 319]

e) Considering desire to have more impactful effect on a person/population, is this study continuous?

- Thank you for the suggestion. The HCHS/SOL is ongoing. Once this data is available, we can expand this analysis to include the third visit. We have added the following sentence to the conclusion: “Future studies can also incorporate data from additional visits of HCHS/SOL.” [Line 364– 635]

7) Comments from Reviewer #3:

a) The authors use an interesting approach here, applying big data and GWAS principles to evaluate neighborhood environment factors associated with T2D. The statistical analyses are sound and conclusions are supported by the reported results. The main weakness of the paper is a lack of discussion regarding selection bias. I would suggest the authors evaluate what the 'general' Hispanic/Latino population looks like in these cities and nationwide, what fraction of them has T2D, neighborhood environment characteristics of areas with Hispanic/Latino population, and how all of this compares to the study populations in the 4 cities. Without this, the generalization of the results are gravely hampered. Even though individual information regarding risk factors will not be available for this evaluation, population level risk factors may offer suggestions as to whether the conclusions from this report are generalizable and in what parts of country (if any).

- Thank you for your comments on considering generalizability. In the US, among Hispanic /Latino adults aged 18 and over the age-adjusted percentage of a diabetes diagnosis in 2022 was 11.5% (CDC, National Diabetes Surveillance System). Non-Hispanic Whites had an age-adjusted percentage of 7.2%. The HCHS/SOL is a multi—center community-based cohort study which target population is all non-institutionalized Hispanic/Latino adults aged 18-74 residing in the four sampling areas. We used sampling weights because participants were selected with different probabilities and to account for non-response by sex, age groups and Hispanic Latino/background. HCHS/SOL is not representative of the Hispanic population in the US. However, based on the 2010 Census, 87% of Hispanic/Latino individuals lived in metropolitan areas of 250,000 people. Our field centers are in cities with large Hispanic/Latino populations. The ranking of these cities among the metropolitan areas in the US with largest Hispanic/Latino populations are New York #1, Chicago #5, San Diego #9, and Miami #11 (US Census Bureau, 2010). We have noted limitations around generalizability in our discussion, including the level of variability in the neighborhood environment in this study. The level of variability in the neighborhood measures in this study population is not likely to reflect the full variability these measures exhibit in the general population, due to the study’s focus on focused geographic areas with high concentrations of Hispanic/Latino residents and low-income households. We have added the following to our limitations: “Lastly, this analysis was focused on 4 major metropolitan areas with large Hispanic/Latino populations in the United States. Results might be different in regions outside the United States or in smaller urban, suburban, or rural areas. Future studies should expand to other cities and regions to increase generalizability.” [Line 353 – 357]

b) The authors' discussion of people who 'moved' between baseline and visit 2 was also confusing. I would suggest they clarify this point further.

- Thank you for your comment. We have re-written this section to clarify. Here is an example of the edited section: “. Differences on several T2D risk factors were observed between those participants who did and did move between visits (S5 Table). This indicates that those who move are not a random subsample and therefore provided a rationale for adjusting the statistical models for moving status.” [Line 297 – 300]

c) Instead of looking for environmental factors that are as strong or stronger than individual factors, I would suggest the authors consider neighborhood environment factors as potential moderators of the relationship between personal risk factors and T2D. This possibility needs to be addressed.

- Thank you for your comments. Effect modification is an important next step. We have added this as a future direction: “Additionally, while investigation of effect modification was out of the scope of this manuscript, it is an important next step for future studies” [Line 308 – 310]

d) Another key factor that deserves discussion is the level of interaction an individual has with their home environment. For instance, some people may interact more with the neighborhood environmental factors at their workplace than home location. This should be considered in Limitations section, along with other personal factors related to physical/mental health, occupation, lifestyle factors, free time, etc. that may influence how a person interacts with their environment in a way that can meaningfully impact their risk of developing T2D.

- Thank you for your comments. We agree that environments outside of one’s residential home are important to factor in and consider. We have added the following after the discussion of the limitations of the 1 km buffer size: “Yet, neighborhood environment outside their 1-km residential area or census tract, such as their place of work and larger community environment, was unaccounted for and may affect T2D risk as well.” [Line 340 – 343]

In addition to addressing the noted concerns we have also edited the following:

1) Earle Chambers’s department of affiliations has been edited to his primary department of Family and Social Medicine.

2) Funding support from New York Regional Center for Diabetes Translation Research (DK111022) has been added for Earle Chambers and Linda Gallo

3) Changed usage of ‘type 2 diabetes’ to ‘diabetes’ in the context of HCHS/SOL studies as ascertainment of diabetes in the HCHS/SOL cohort was not able to distinguish between type 1 and type 2 diabetes. However, given the age of the HCHS/SOL cohort, the greater majority of incident cases are likely type 2. We have noted this in the methods and have added it as a limitation in the discussion. However, we still used type 2 diabetes when discussing the potential risk factors and citing references that used type 2 diabetes.

Thank you again for all the feedback. A marked-up and clean version of the edited manuscript has been uploaded.

Sincerely,

Cara M Smith

---

## [Decision Letter · Decision Letter 1]

Neighborhood Environment and Incident Diabetes, A Neighborhood Environment-Wide Association Study (‘NE-WAS’): Results from the Hispanic Community Health Study/Study of Latinos (HCHS/SOL)

PONE-D-24-39866R1

Dear Dr. Smith,

We’re pleased to inform you that your manuscript has been judged scientifically suitable for publication and will be formally accepted for publication once it meets all outstanding technical requirements.

Kind regards,

Kamran Baig, MPH, MBBS

Academic Editor

PLOS ONE

Additional Editor Comments (optional):

Thank you for your thoughtful revisions to your manuscript. This have further enhanced the clarity and scientific rigor.

Reviewers' comments:

Reviewer's Responses to Questions

**Comments to the Author**

Reviewer #2: All comments have been addressed

Reviewer #3: All comments have been addressed

2. Is the manuscript technically sound, and do the data support the conclusions?

Reviewer #2: Yes

Reviewer #3: Yes

3. Has the statistical analysis been performed appropriately and rigorously?

Reviewer #2: Yes

Reviewer #3: Yes

4. Have the authors made all data underlying the findings in their manuscript fully available?

Reviewer #2: Yes

Reviewer #3: Yes

5. Is the manuscript presented in an intelligible fashion and written in standard English?

Reviewer #2: Yes

Reviewer #3: Yes

Reviewer #2: Previous comment are well addressed. With the review done, the knowledge value of the manuscript has been enhanced. Looking forward to future outcomes from the study.

Reviewer #3: Authors have done a commendable job with the revisions. I believe it is improved because of their efforts. I have no further comments on this manuscript.

**Do you want your identity to be public for this peer review?** For information about this choice, including consent withdrawal, please see our Privacy Policy

Reviewer #2: No

Reviewer #3: No

---

## [Editor Report · Acceptance letter]

PONE-D-24-39866R1

PLOS ONE

Dear Dr. Smith,

I'm pleased to inform you that your manuscript has been deemed suitable for publication in PLOS ONE. Congratulations! Your manuscript is now being handed over to our production team.

Kind regards,

on behalf of

Dr. Kamran Baig

Academic Editor

PLOS ONE